# Planar Junctionless Field-Effect Transistor for Detecting Biomolecular Interactions

**DOI:** 10.3390/s22155783

**Published:** 2022-08-02

**Authors:** Rajendra P. Shukla, J. G. Bomer, Daniel Wijnperle, Naveen Kumar, Vihar P. Georgiev, Aruna Chandra Singh, Sivashankar Krishnamoorthy, César Pascual García, Sergii Pud, Wouter Olthuis

**Affiliations:** 1BIOS Lab-on-a-Chip Group, MESA+ Institute for Nanotechnology, Max Planck Center for Complex Fluid Dynamics, University of Twente, P.O. Box 217, 7500 AE Enschede, The Netherlands; j.g.bomer@utwente.nl (J.G.B.); d.wijnperle@utwente.nl (D.W.); w.olthuis@utwente.nl (W.O.); 2Device Modelling Group, School of Engineering, University of Glasgow, Glasgow G12 8LT, UK; naveen.kumar@glasgow.ac.uk (N.K.); vihar.georgiev@glasgow.ac.uk (V.P.G.); 3Nano-Enabled Medicine and Cosmetics Group, Materials Research and Technology Department, Luxembourg Institute of Science and Technology (LIST), L-4362 Belvaux, Luxembourg; aruna.singh@list.lu (A.C.S.); sivashankar.krishnamoorthy@list.lu (S.K.); 4Nanoscale Engineering for Devices & Bio-Interfaces, Nanotechnology Unit of the Materials Research and Technology Department, Luxembourg Institute of Science and Technology (LIST), L-4422 Belvaux, Luxembourg; cesar.pascual@list.lu

**Keywords:** planar junctionless FETs, pH sensor, proteomics, peptidomics, peptide-protein interaction, therapeutics, diagnostics

## Abstract

Label-free field-effect transistor-based immunosensors are promising candidates for proteomics and peptidomics-based diagnostics and therapeutics due to their high multiplexing capability, fast response time, and ability to increase the sensor sensitivity due to the short length of peptides. In this work, planar junctionless field-effect transistor sensors (FETs) were fabricated and characterized for pH sensing. The device with SiO_2_ gate oxide has shown voltage sensitivity of 41.8 ± 1.4, 39.9 ± 1.4, 39.0 ± 1.1, and 37.6 ± 1.0 mV/pH for constant drain currents of 5, 10, 20, and 50 nA, respectively, with a drain to source voltage of 0.05 V. The drift analysis shows a stability over time of −18 nA/h (pH 7.75), −3.5 nA/h (pH 6.84), −0.5 nA/h (pH 4.91), 0.5 nA/h (pH 3.43), corresponding to a pH drift of −0.45, −0.09, −0.01, and 0.01 per h. Theoretical modeling and simulation resulted in a mean value of the surface states of 3.8 × 10^15^/cm^2^ with a standard deviation of 3.6 × 10^15^/cm^2^. We have experimentally verified the number of surface sites due to APTES, peptide, and protein immobilization, which is in line with the theoretical calculations for FETs to be used for detecting peptide-protein interactions for future applications.

## 1. Introduction

Recent developments in peptidomics and proteomics have enabled the rapid progress of novel personalized therapies [1,2]. Peptides are short sequences of amino acids with high specificity and affinity towards binding targets [3,4]. Some of them represent protein epitopes that carry diagnostic and therapeutic information as their interaction with the major histocompatibility complex (MHC proteins) can determine a patient’s specific response to a possible vaccine (e.g., for cancer) [5,6]. Screening such sequences for their interaction with antibodies and MHC proteins is of great interest in modeling the response of the immune system. However, the intrinsic variability of these peptide sequences hinders high throughput screening to cover all possible combinations of amino acids [7]. Transducing such interactions into readable signals requires a multiplexed setup of label-free immunosensors that allows detection in the physiological range [5,8,9]. Current sensing technologies have limited multiplexing capabilities and require labelling of the molecules (e.g., ELISA) [9]. Therefore, there is a need for a multiplexed setup with controlled immobilization of these peptide sequences on devices that enable highly sensitive and label-free sensing of the target analytes. Field-effect transistor (FET)-based immunosensors are good candidates for multiplexed label-free sensing due to their high scalability, compatibility with current CMOS technology, fast response time, and label-free sensitivity [10,11,12,13]. When functionalized with short sequences of peptides, a FET gate can detect the binding of proteins in close proximity to the sensitive region within the Debye screening length of the protein solution [14]. Nanowire-based FETs are one of the most highly investigated structures among them because of the 3D gating effect and faster mass transport towards the sensing area [15,16,17,18]. The surface area-to-volume ratio allows the adsorption of the analytes in 2D as compared to planar adsorption. However, nanowire devices are still facing several challenges in clinical applications due to reliability issues [19]. Here, we propose a simpler design in terms of fabrication point-of-view, called planar junctionless FETs, where the conducting channel acts as a resistor and the carrier density in the channel resistor can be modulated by applying the gate voltage by means of a reference electrode in a given pH of the electrolyte solution [20]. The advantage of this device lies in its relative simplicity; it does not require the fabrication of shallow implanted p-n junctions in the source and drain areas. Moreover, the planar structure of the device allows more robust functionalization of the sensing surface as compared to any other non-planar structure [17]. We have used lightly doped thin device layer SOI wafers to demonstrate their suitability for detecting small changes in charge at the electrolyte-oxide surfaces (i.e., caused by the interaction of the proteins with peptides immobilized on the gate surface, which is the long-term goal of this work). A larger planar surface area allows a better signal-to-noise ratio and also less stringent requirements to counter reliability issues, e.g., from pin-holes, as compared to the nanowire counterparts [21].

Figure 1 shows the schematic of the proposed device design (Figure 1A) and the cross-sectional view of the device layout (Figure 1B). We have overcome several fabrication related challenges during the process. For example, ohmic contact with the lightly doped thin device layer is best achieved by the formation of a thin layer of PtSi alloys at the interface of Ta/Pt and silicon. However, this process is too sensitive to the thickness of the silicide formation and the annealing temperature to provide reproducible results with our device layer thickness [22,23,24,25,26,27,28,29]. We have overcome this issue by optimization of the annealing process to get a reliable planar junctionless FET device, and in the end, we have demonstrated the pH sensing performance of our fabricated device. Theoretical modeling and simulation were done using experimental data to calculate the surface states and charge density present at the oxide layer. Further, we have experimentally calculated the number of surface sites after silanization of the gate oxide surface with APTES, peptide, and protein functionalization. These data will be used for optimizing future devices where the oxide surface of the FETs will be functionalized with different chemistries. We have tested the stability of the device over time (drift analysis) and confirmed its suitability for future application as a label-free sensor of peptide–protein interactions. We anticipate that the proposed rational device design can be an optimal solution for reproducible multiplexed sensing of peptide–protein interactions [30,31].

## 2. Materials and Methods

### 2.1. pH Test Buffers

Tetrabutylammonium chloride (TBACl), tetrabutylammonium hydroxide solution (TBAOH), acetic acid, boric acid, and orthophosphoric acid were purchased from Merck (Sigma Aldrich). As a background electrolyte, 0.1 M of TBACl was used. First of all, a universal buffer mixture (UBM) was prepared by mixing 0.5 M acetic acid, 0.5 M boric acid, and 0.5 M orthophosphoric acid. To buffer the solution, 200 µL of 0.5 M UBM was mixed with 50 mL of 0.1 M TBACl. The pH at the start was around 2.7 at 25 °C. Titration was performed with 0.1 M TBAOH in 20 steps of 0.4 mL and the pH at the end was found to be around 10.5. Back titration was performed with 0.1 M HCl.

### 2.2. Design Considerations for Planar Junctionless FETs

#### 2.2.1. Wafer Specifications

SOI wafers were purchased from IceMOS Technology, Ltd. with a diameter of 100.00 ± 0.20 mm, device orientation <100> ± 1.0 degree, silicon device layer thickness of 2.00 ± 0.50 µm, and p-type device layer resistivity of 1–10 Ohm.cm were used for the fabrication of planar junctionless FETs. 

#### 2.2.2. Thin and Lightly Doped Device Layer

The thin and lightly doped silicon device layer is required to have a higher sensitivity with a high on/off drain current ratio. For this purpose, the device layer was thinned down using successive wet oxidation and etching of the SiO_2_ layer. We have fabricated FET devices with a device layer thickness of 250–300 nm. 

### 2.3. Device Fabrication

The fabrication of the planar junctionless FETs consists of the following steps as depicted in Figure 2.

#### 2.3.1. Cleaning SOI Wafer

The process started with the pre-furnace cleaning of SOI wafers in 99% HNO_3_ for 10 min to remove organic traces, followed by rinsing in DI water for the removal of traces of chemical agents. The rinsed SOI wafers were further cleaned in 69% HNO_3_ at 95 °C for 10 min to remove metallic traces. The wafer was further rinsed in DI water and dried with nitrogen. To remove the native oxide, the wafer was transferred to a 1% HF etching chamber at room temperature. Within several seconds, the surface became hydrophobic, which is an indication of the removal of native oxide from the surface. The wafer was further rinsed in DI water and dried with nitrogen, which was then loaded into the wet oxidation furnace (step (i) in the process flow).

#### 2.3.2. Thinning Silicon Device Layer

The first wet oxidation was done at 1150 °C for 15 h to obtain a thickness of 2.6 µm. The oxide was etched in a 50% HF solution for approximately 3 min until the surface became completely hydrophobic. Similar to this, the second step of oxidation to get an oxide thickness of 1.1 µm was done for 3 h at 1150 °C. This 1.1 µm thick oxide layer was thinned down to a 300 nm oxide layer by etching in buffered HF acid solution to be used as a mask for doping the source and drain regions (steps (ii) and (iii) in the process flow).

#### 2.3.3. Doping Source and Drain Regions

The source and drain regions were first opened by photolithography. This process started with HMDS priming on a spin coater at 4000 rpm for 30 s. The wafer was further spin coated with photoresist Olin OiR 907-17 at 4000 rpm for 30 s, followed by pre-baking at 95 °C for 90 s to remove the residual solvent from the resist film after spin coating. The spin-coated wafer was exposed to UV-LED light with an exposure dose of 100 mJ/cm^2^. The exposed wafer was then post-exposure baked at 120 °C for 60 s on a hot plate. The wafer was developed for 60 s, followed by rinsing in DI water and drying with nitrogen. The patterns were inspected using an optical microscope. The developed wafer was then baked at 120 °C for 10 min, followed by UV-ozone cleaning for 5 min to remove any residue of the photoresist. The patterned oxide layer was etched in BHF solution, and the resist was stripped in HNO_3_. Boron doping was done using Plasma Enhanced Chemical Vapor Deposition of a 100 nm boron doped oxide, covered with a 250 nm undoped capping oxide layer, followed by drive-in at 1100 °C for 30 min. The oxide was then removed in a 50% HF solution (steps (iv) and (v) in the process flow).

#### 2.3.4. Defining Silicon Islands

Next, silicon nitride was deposited to act as a mask to define the silicon islands. The silicon islands were defined using photolithography. The nitride was then removed by dry etching and, subsequently, the silicon was etched in TMAH at 70 °C [32]. The color change was observed as proof of the complete etching of silicon. Next, the nitride layer on top of the silicon islands was removed by etching it in a phosphoric acid solution at 180 °C for 10 min (steps (vi), (vii), and (viii) in the process flow).

#### 2.3.5. Source and Drain Patterning

Next, 10 nm of gate oxide was grown on the islands by dry oxidation for 25 min at 900 °C before defining the source/drain area. Source and drain regions were defined using another photolithography step followed by BHF etching of oxide (steps (ix) and (x) in the process flow).

#### 2.3.6. Metal Contacts Lift-Off

Metal lift-off patterns were defined using photolithography on double layer photoresist: LOR5A and Olin OiR 907-17. Ta/Pt of 2 nm/100 nm was sputtered and lift off in acetone solution with an ultrasonic bath [33]. The LOR5A photoresist was then removed in a 99% HNO_3_ solution, followed by rinsing in DI water and drying with nitrogen. The Ta/Pt patterned wafer was then annealed at 350 °C for 10 min to improve the electrical contact as it reduces the interface trap density at the metal-semiconductor interface. Although the forming of a thin layer of PtSi alloy after annealing is supposed to ensure ohmic contacts between metal leads and source/drain regions, it is, in practice, rather challenging due to the sensitivity of the PtSi formation to the annealing temperature [23,34]. Moreover, the thin device layer makes the process of annealing prone to irreproducibility because annealing time and temperature can considerably affect the thickness of the device layer due to its being consumed during PtSi formation. To address these challenges and to ensure robustness of the fabrication process, the source and drain regions were doped (see Appendix A for more details) [step (xi) in the process flow].

#### 2.3.7. SU-8 Passivation Layer and Channels

SU-8 patterns were defined in SU-8-2005 (thin SU-8 layer opening at gate and contact area) and SU-8-100 (thick layer, SU-8 channels) using photolithography. First, the SU-8 layer was spin coated at 500 rpm for 10 s (step I), and then at 5000 rpm for 30 s (step II). The spin coated wafer was soft baked at 95 °C for 2 min. The wafer is then exposed at 90 mJ/cm^2^ using UV-LED. The exposed wafer was then post exposer baked at 95 °C for 2 min. The wafer was then developed in RER600 developer for 1 min, followed by rinsing in isopropanol and drying with nitrogen. The SU-8 channels were defined using another lithography step. First, the SU-8 layer was spin coated at 500 rpm for 10 s (step I), and then at 3000 rpm for 30 s (step II). The spin coated wafer was soft baked at 95 °C for 15 min. The wafer was then exposed at 400 mJ/cm^2^ using UV-LED. The exposed wafer was then post exposure baked at 95 °C for 10 min. The wafer was developed in RER600 developer for 10 min, followed by rinsing in isopropanol and drying with nitrogen. The wafer was then hard baked at 135 °C for 30 min before being diced into chips. One wafer consists of several chips with test patterns and junctionless FETs. Therefore, the wafer was diced into chips using a dicing saw (Disco DAD3220) before use [steps (xii) and (xiii) in the process flow].

### 2.4. Chip Design and Encapsulation

The planar junctionless FET chip was designed using CleWin software for a 100 mm wafer mask. The dimension of a single chip is 1 × 1 cm^2^, which consists of 15 metal contact pads and three microfluidic channels (Figure 3A). Figure 3B shows SU-8 microfluidic channels. There are 12 FET devices in total (four devices inside each microfluidic channel) with a common source along with a pseudo reference electrode available in this chip. The channel length and width of the device are 4 µm and 12 µm. Figure 3C shows a single junctionless FET device with an open gate area. The pseudo reference electrode has three terminals for each microfluidic channel, which are supposed to be electroplated with silver/silver chloride in future applications. In this work we have used an external silver/silver chloride reference electrode for the simplicity of the measurement set-up. This three-channel based design is adopted by considering a long-term goal of capturing biomolecular interactions where these FETs will be functionalized with different sequences of peptides and their interactions with proteins will be tested. The diced chips were wire bonded for electrical connections to a PCB and insulated using epoxy glue. For proper insulation and hardening of the epoxy glue on the chip, the PCB with epoxy glue was heated on a hot plate for 2 h (Figure 3D). After that, the PCB connected chip was cleaned using plasma for 5 min. Prior to the pH characterization of these devices, the leakage test of the PCB-connected device was performed by putting the device in water and buffer solution and connecting it to the power supply. No leakage current between different electrodes was observed over several hours, which is indicative of the proper insulation of the device with epoxy glue. A microscopic inspection was done to make sure that there was no water that leaked through the SU-8 layer. After having a detailed test of the devices, the pH characterization was done.

### 2.5. pH Measurement Setup

The encapsulated device was submerged in the buffer solution along with a reference electrode (REF201, red-rod reference saturated in 3M KCl solution, Radiometer Analytical,) with a connection to the source meter. The measurement started in a mixture solution of 200 µL of 0.5 M UBM and 50 mL of 0.1 M TBACl (pH 2.7). The pH of the solution was changed by adding 400 µL of 0.1M TBAOH in steps followed by stirring the solution to make it homogeneous mixture. After stabilization, the pH was measured before recording the pH response. The pH meter (Mettler-Toledo B.V., S-400 basic) was used to measure the pH of the solution, and it was calibrated before use.

### 2.6. V_gs_ vs. pH and Ids vs. pH Characterization

First, the drain current, I_ds_, was measured as a function of drain-source voltage, V_ds_, and gate voltage, V_gs_. These characterizations were done at a fixed pH to find out the set-point for the device operation. After establishing the optimal setpoint, the V_gs_ was measured as a function of pH as well as I_ds_ to show the sensitivity of the device towards pH change. The drift characterizations were done using the same set-point. For the voltage sensitivity analysis, the V_gs_ was adjusted to maintain the constant I_ds_ for every pH of the electrolyte solution. V_ds_ was kept at 0.05 V. For current sensitivity analysis, the I_ds_ was recorded for a varying pH of the electrolyte solution at a V_gs_ and V_ds_ of −0.5 V and 0.05 V. The voltage and current sensitivities were calculated from the linear fit of V_gs_ vs. pH and Ids vs. pH characteristics. 

### 2.7. Drift Analysis 

I_ds_ was recorded for 100 min with a varying pH every 10 min and then for 2 h at a constant pH to check the drift over time. Current drift over a time period of 2 h was calculated by subtracting the current value at the start and after 2 h at a constant pH. The pH drift over time was calculated using the current drift over time at a fixed pH and the current sensitivity of the pH response.

## 3. Results and Discussion

### 3.1. pH Characterization of 2D Planar JUNCTIONLESS FETs

Before we started the pH characterization of the FETs, the set-point (or working point) of the device was decided such that the device operates in a linear region of operation to include the ohmic contribution in the current variation. To decide the set-point of the junctionless FETs for pH characterization and sensitivity analysis, we measured the I_ds_ vs. V_ds_ and I_ds_ vs. V_gs_ characteristics in a constant pH solution of 4.91 (Figure 4A,B). These characterizations provide the working voltage range (set-point) for these devices, which is V_gs_ = −0.5 V and V_ds_ = 0.05 V.

Next, the device was characterized for the voltage and current sensitivities as a function of the pH. The surface potential is changed by V_gs_ and pH, and V_gs_ is related to the threshold voltage V_th_ [35]. Therefore, the shift in the V_th_ is observed as a change in V_gs_. As the pH changes from acidic to basic, the shift in the V_th_ moves towards a less negative value. The shift of the V_th_ with increasing pH must be compensated for by increasing V_gs_ to keep the concentration of carriers in our p-type channel the same. This can further be detailed by the relationship between the surface potential and the pH, which is derived by combining the electrostatic interactions at the dielectric surface and the distribution of ions inside the electrolyte (Equation (1)) [35].
(1)$$\frac{\partial \psi_{0}}{\partial pH_{B}}=2.303\frac{k_{B}T}{q}\alpha$$
where *ψ*_0_, *pH_B_*, *k_B_*, *T* and *q* represent the surface potential, bulk pH of the electrolyte, the Boltzmann constant, the absolute temperature, and the elementary charge, respectively. *α* is a sensitivity parameter with a value varying between 0 and 1, depending on the intrinsic properties of the oxide. For *α* = 1, the sensor shows maximum sensitivity called Nernstian sensitivity which is 59.2 mV/pH at 298 K.

For voltage sensitivity analysis, the change in the V_gs_ (as a result of the shift in V_th_) was recorded for a constant I_ds_ as a change in pH at a V_ds_ of 0.05 V. Figure 4C shows the variation of V_gs_ for different pH values at a fixed V_ds_ of 0.05 V for constant currents of 5 (red circles), 10 (blue circles), 20 (green triangles), and 50 nA (purple squares). From the V_gs_ vs. pH characteristics, the sensitivities were calculated from the linear fitting. It is found that for constant currents of 5, 10, 20, and 50 nA, voltage sensitivities are 41.8 ± 1.4, 39.9 ± 1.4, 39.0 ± 1.1, and 37.6 ± 1.0 mV/pH at a fixed drain to source voltage of 0.05 V. This shows that our junctionless FET devices with 10 nm of SiO_2_ are sensitive to pH change, as expected and reported in the literature [36]. The slight change in voltage sensitivity values for different constant current values is due to the dominant effect of noise current levels at lower constant current values. The sensitivity (α) is calculated using equation 1. The average value of the sensitivity factor (α) calculated for all the constant currents was found to be 0.70 ± 0.03. The calculated sensitivity value is in good agreement with the literature values for pH response at the SiO_2_ surface [37,38]. The calculated current sensitivity from the I_ds_ vs pH characteristics was found to be 38.9 ± 2.1 nA/pH, with a wide range of current response (50 to 400 nA; Figure 4D). This wide current range provides us with an insight of almost complete depleted channel to a fully conducting channel. The current sensitivity was further used to calculate the stability of the sensor for pH change in drift analysis. We have plotted the I_ds_ vs. charge density using an analytical model which shows a sensitivity for change in charge density of 0.20 ± 0.01 nA/C/cm^2^ (see Appendix A for more details). For a positively charged surface on the oxide electrolyte interface, the channel is almost closed, and a minimum current is observed. As the charge density changes to a negatively charged interface, we observe the small change of that charge effect in terms of drain to source current. These results show promising proof-of-concept device characteristics to be used for sensing interactions of biological molecules, which is one of the long-term project goals with chemistries that can generate different charge densities due to different surface sites on the surface [39,40].

### 3.2. Calculating Surface States

To evaluate the nature of our surface oxide, we have used the site-binding model with the Gouy–Chapman–Stern (GCS) model. Matlab (R2022a) has been used for analytical modeling and simulation of the experimental data and to calculate the surface states present on the oxide layer. Using the linear regime of our sensors, we can obtain the number of silanol groups that exchange protons with the electrolyte, and thus contribute to the sensitivity, providing a good value of the quality of our oxide. We have considered a stern capacitance of 0.8 F/m^2^. Equating the site-binding model with the Gouy–Chapman–Stern theory for the double layer capacitance provides an equation of the 5th order, which results in possible saturation at a higher pH range due to an approximation of 4*kb/ka*⪡1 with unwanted ripples even with the iterative method solution of the 5th order equation [39,40,41]. Thus, we solved both equations independently with an assumption of the same surface potential values and then equated them later on with a tolerance of 10–50 as explained in detail in [40]. Such an approach is more accurate and flexible enough to be used for all types of oxides or even at the surface with more than two affinity sites with different dissociation constants. In this approach, zeta potential was indirectly considered as an experimental index of the surface states by correlating the zeta potential to the surface potential with a potential drop across the stern layer. As per the Gouy–Chapman–Stern model, the stern layer (uncharged dielectric) between the diffuse layer and the oxide–electrolyte interface decreases the effective potential at the shear plane (zeta potential). Figure 5A shows the scheme of surface cites present in SiO_2_. The electrolyte concentration and device parameters were kept the same as in the experimental setup. Figure 5B shows the calibration of the simulated model with the experimental data in terms of reference gate bias with respect to the electrolyte pH. The graphs for different current values of 5 nA, 10 nA, 20 nA, and 50 nA have been plotted while representing the possible root mean square error (RMSE) using surface states as the fitting parameter. The following equations were used to calculate different parameters in the model.
(2)$$\sigma_{DL1}=qN_{S}\left(\frac{cH_{s}^{2}-K_{a}K_{b}}{K_{a}K_{b\;}+K_{b}cH_{S\;}+cH_{S}^{2}}\right)\;\;\;$$
(3)$$cH_{s}=cH_{B}exp\frac{-\Psi_{0}}{2V_{T}},\;cH_{B}=10^{-pH_{B}}\;\;$$
(4)$$\Psi_{0}=\Psi_{stern}+\Psi_{\xi }=\frac{Q_{0}sinh\left(\Psi_{\xi }/V_{T}\right)}{C_{stern}}+\Psi_{\xi }$$
where $\sigma_{DL1}\;$is the surface charge density, $N_{S}$ is the number of surface states, $\;K_{a}=10^{-{\mathrm{pK}}_{a}}$ and $K_{b}=10^{-{\mathrm{pK}}_{b}}$ are the dissociation constants, $cH_{s}$ is the surface proton concetration, $cH_{B}$ is the bulk proton concentration, $\Psi_{0}$ is the surface potential, $V_{T}$ is the thermal voltage, $\Psi_{stern}$ is the potential drop across the stern layer and $\Psi_{\zeta }$ is the potential drop across the diffuse layer. We have used the dissociation constants reported in the literature for silanol groups: pK_a_ = 6 and pK_b_ = −2 [42]. Keeping the constant affinity of the silanol sites, the density of surface states is varied as a fitting parameter. A surface potential-pH curve is extracted by self-consistently solving the site-binding and GCS models. As an assumption, the curves (four samples for different current values) are supposed to have the same potential near the isoelectric point (pH = 2) and it was considered a starting point to decide the slope of the curve. Every sample was compared with each simulated surface potential-pH curve for different surface states, and the RMSE was calculated. The closest curve to the corresponding sample was extracted with the minimum RMSE value. Assuming constant affinity values, the possible induced doping for different current values may be the reason behind the variation of surface states that can be counted as an error. The obtained mean value of the number of surface states is 3.8 × 10^15^/cm^2^ with a standard deviation of 3.6 × 10^15^/cm^2^. The obtained value of the surface potential used to get these surface states is in good agreement with the simulated and experimental work [40]. Such a high value of surface states signifies the quality of deposited oxide, resulting in high sensitivity for SiO_2_ FETs. These values for the number of surface sites were further used to compare the surface sites due to different surface functionalization in the next sections.

### 3.3. Measuring APTES Functionalization and Monitoring Peptide-Protein Interactions Using Surface Plasmon Resonance (SPR)

Gold SPR chips with a silicon oxide coating (Au/SiO_2_) were used to study peptide-protein interactions [43,44]. The use of the SiO_2_ surface allows mimicking the conditions to be encountered on the silicon oxide surface of the FET sensors (for more information about the APTES functionalization protocol and steps, see Appendix A. Surface functionalization of SiO_2_ with APTES, peptides, and proteins). The verification of the APTES layer has been done by XPS analysis on bare SiO_2_ and silanized (APTES coated SiO_2_ chip) (see Appendix A: Surface characterization using XPS). The molecular densities at different steps of functionalization are presented in Table 1. The APTES density on the sensor surface was found to be 1.3 × 10^15^/cm^2^, which corresponds to the presence of a monolayer of APTES. The peptide layers with non-specific antibodies showed no antibody retention on the surface, confirming the specificity of the sensor to the specific peptide antibody interactions. The experimental value of APTES density is comparable with the surface sites present in SiO_2_ calculated using an analytical model showing complete coverage of oxide surface sites after functionalization. The density of the peptides and proteins is much lower than the number of silanol groups/APTES groups. Based on the SPR experiments to measure the number of peptide protein interactions, we expect a significant sensitivity that at least will be equivalent to the number of neutralized peptides interacting with proteins. (Table 1).

### 3.4. Stability of the Sensor

Figure 6A shows the drain-to-source current, I_dS_, vs. time at different pH values. The gate-to-source voltage, V_gS_, and the drain-to-source voltages, V_dS_, were fixed at −0.5 V and 0.05 V. From Figure 6A, it can be clearly seen that the device is responsive to the pH change happening at the dielectric–electrolyte interface. As the pH increases, the OH-concentration in the solution increases and that is why the charge carriers in the channel regions for a p-type channel also increase and that is why there is an increase in the current. The measurement for each pH was recorded for 10 min and after changing the pH, the solution was stirred to mix the ions and make a homogenous solution for the pH measurement. The short time for stabilization restricts the charging of the electrical double layer due to higher screening with increased ion concentration, resulting in a drift in the response.

Figure 6B shows the drain-to-source current vs. time for several pH values. The response of the device was recorded for a time period of 2 h for each pH value (3.43, 4.91, 6.84, and 7.75). We have calculated the drift over time, which shows −18 nA/hour (pH 7.75), −3.5 nA/hour (pH 6.84), −0.5 nA/h (pH 4.91), and 0.5 nA/h (pH 3.43) for the corresponding drift in pH value over time calculated using the current sensitivity obtained from Figure 4D of −0.45, −0.09, −0.01, and 0.01 per hour. It can be seen that the device has a stable response at lower pH values with a small drift, which is expected due to the fluctuation in the electric field because of the larger surface area of the device. The response at a higher pH (7.75) value takes time to stabilize as some of the mobile charges at the dielectric surface take time to charge/discharge at the surface. That is why the drift is observable for a longer time as compared to lower pH values. From this analysis, it is clear that working at a low pH value has a gain in stability at the cost of sensitivity. We have used these sensors for several days and they have shown stable sensitivity. We have tested the response of this device in phosphate buffered saline (PBS) as well. It is found that the drift is −6 nA/h for pH 7.74, which is less than the buffer used in Figure 6B (Please see Appendix A for more details). This signifies that the drift is buffer solution dependent as the ion concentration is different and it can be minimized by using an appropriate buffer solution. The calculated values of drift (% ΔI_ds_/h = 6% for TBACl and 0.2% for phosphate buffer solution) at pH 7.8 are below the change in the current observations for detecting the interaction of biological molecules (e.g., protein–protein [15,38,45]), which shows that our proposed device is suitable for such measurements.

## 4. Conclusions

Here, we have presented the fabrication and characterization of junctionless FETs. We have optimized different process parameters, e.g., annealing temperature and time, to achieve ohmic contacts in these devices. To this end, we have doped the source and drain regions. The fabricated device has shown the expected voltage and current sensitivity for pH measurements as per the literature. Theoretical simulation and modeling have shown that the calculated number of surface sites of an oxide surface is comparable with the experimentally obtained results of the APTES surface functionalization. Further, the peptide and protein surface density were calculated using SPR experiments, which shows that the numbers of peptides and proteins are very close; therefore, we expect a minimum significant sensitivity. Further, the stability of the device was tested using drift analysis that shows stability of the device in the range required for detecting peptide–protein interactions. This rational design of junctionless FET chips will later be used as a multiplexed set-up of immunosensors to detect the interaction between proteins, which is selective for different peptide sequences functionalized on the sensor surface [46]. Later on, these devices will be integrated into a microfluidic setup with a more automated setup for detecting peptide and protein interactions.

## Figures and Tables

**Figure 1 sensors-22-05783-f001:**
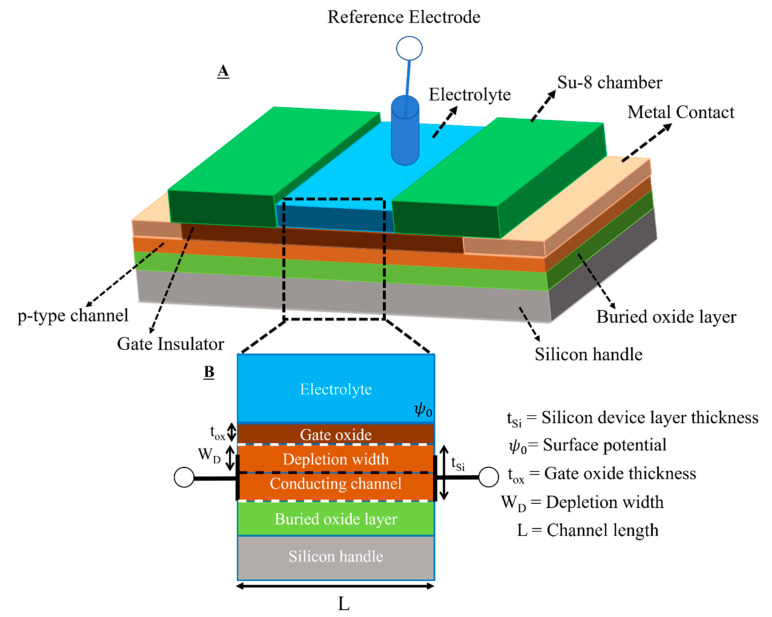
(**A**) Schematic of the proposed planar junctionless FETs and (**B**) cross-sectional view of the proposed device design.

**Figure 2 sensors-22-05783-f002:**
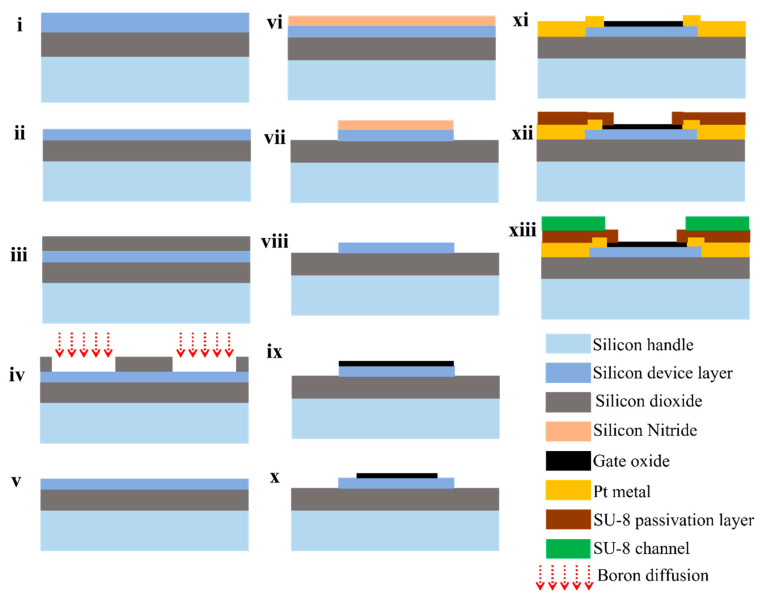
Fabrication process flow of the planar junctionless FETs. (**i**) cleaning SOI wafer, (**ii**) thinning of device layer using oxidation and etching, (**iii**) silicon dioxide growth as a masking layer, (**iv**) patterning silicon dioxide and boron diffusion, (**v**) etching silicon dioxide, (**vi**) silicon nitride deposition, (**vii**) patterning silicon nitride to define silicon islands, (**viii**) etching silicon nitride, (**ix**) gate oxide growth, (**x**) patterning gate oxide to define source and drain regions, (**xi**) Ta/Pt Metal lift-off, (**xii**) patterning SU-8 passivation layer, and (**xiii**) patterning SU-8 channels.

**Figure 3 sensors-22-05783-f003:**
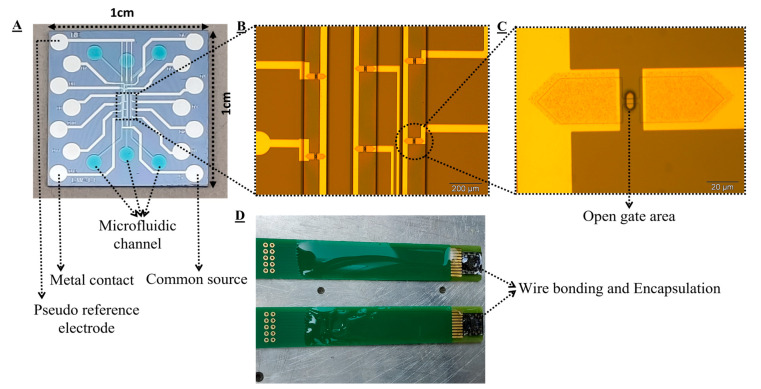
Fabricated chip design and encapsulation. (**A**) image of the single chip, (**B**) SU-8 channels, (**C**) gate opening of a single FET in a thin SU-8 layer. Color change is observed at doped source and drain regions due to the formation of a PtSi alloy, and (**D**) encapsulation with epoxy glue.

**Figure 4 sensors-22-05783-f004:**
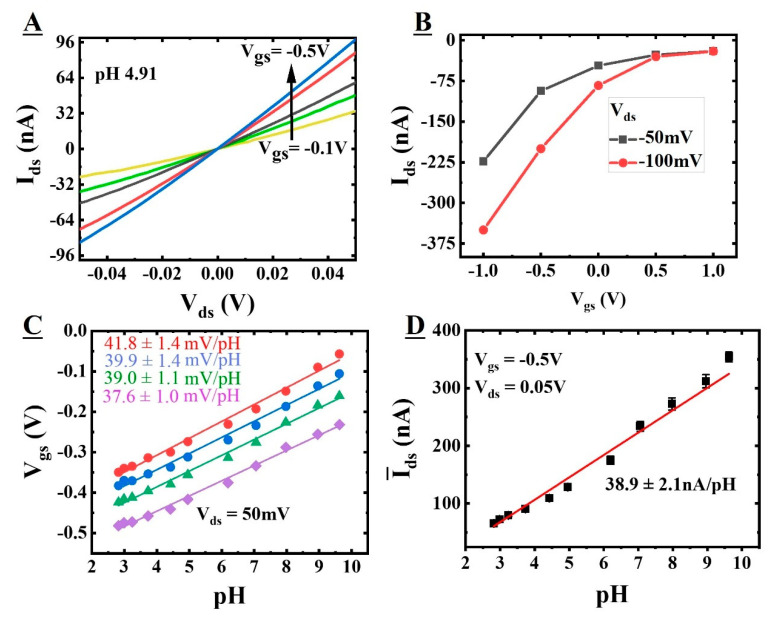
I-V characteristics at fixed pH of 4.91 (**A**) I_ds_ vs. V_ds_ for a varying gate voltage, applied via the reference electrode (−0.1 to −0.5 V in steps of −0.1 V) with V_ds_ ranging from −0.05 V to 0.05 V, and (**B**) I_ds_ vs. V_gs_ for input gate voltage range of −1 to 1 V, applied via the reference electrode for a V_ds_ of 0.05 and 0.1 V. Voltage and current sensitivity analysis (**C**) V_gs_ vs. pH, and (**D**) average I_ds_ vs. pH.

**Figure 5 sensors-22-05783-f005:**
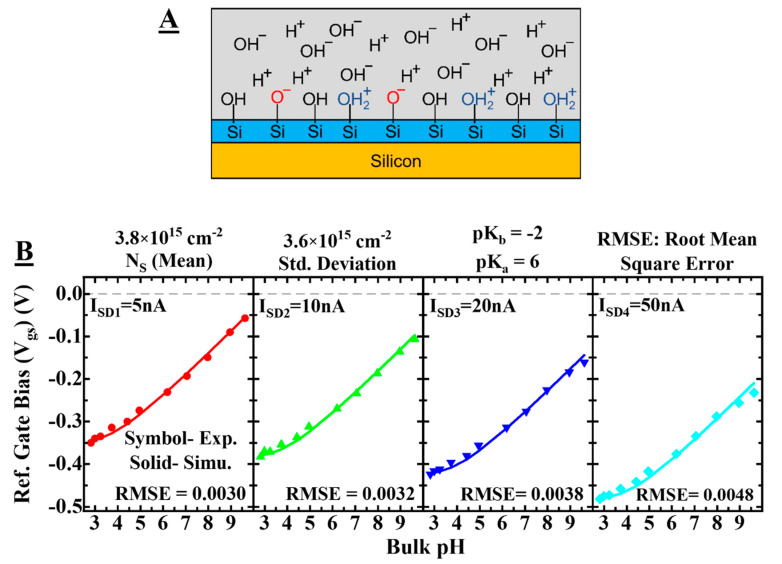
Simulation and modeling of the junctionless FETs. (**A**) Scheme for the surface sites available in SiO_2_. (**B**) Calibration of the simulated model with the experimental data in terms of reference gate bias with respect to the electrolyte pH. Separated graphs for different current values of 5 nA, 10 nA, 20 nA, and 50 nA while representing the possible RMSE error using surface states as the fitting parameter. The used color codes are the same for corresponding current values.

**Figure 6 sensors-22-05783-f006:**
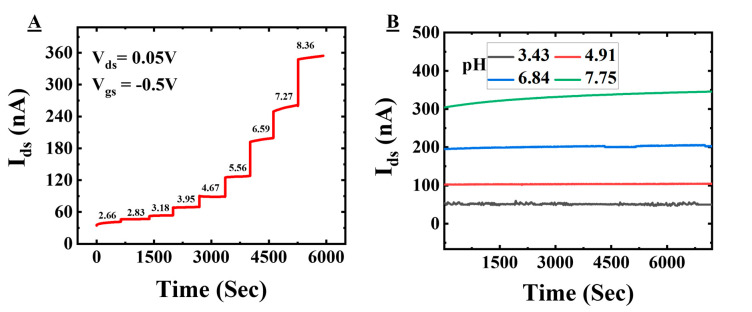
Drift characterization. (**A**) Current vs time step response for different pH and (**B**) current vs. time for a longer time for several constant pH values.

**Table 1 sensors-22-05783-t001:** Surface molecular densities with respect to the surface mass absorption.

Surface Groups	Surface Concentration (ng/cm^2^)	Calculated Molecular Density (/cm^2^)
APTES	470	1.3 × 10^15^
Peptide	70	2.3 × 10^13^
*p*Ab (solution concentrations in the range of 0.1–20 µg/mL)	7–934	2.8 × 10^10^–1.8 × 10^12^

## Data Availability

The data presented in this study are available on request from the corresponding author.

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
