# Peer review of "Planar Junctionless Field-Effect Transistor for Detecting Biomolecular Interactions"

_sensors, 2022, doi:10.3390/s22155783_

Round 1

Reviewer 1 Report

Very good paper.Only 2 improvements requested

Why zeta potential was not considered as experimental indes of the surface state?

There is novcorrespondance along all the paper between sensitivi and precisione data expressed by diagrams,tables or data

I suggest also to improve also the introduction:pH is the obvious parameter involved in.biomolecules interaction so yhat at the beginning the paper risks to be considered of modest intersst

Reviewer 2 Report

I have reviewed the manuscript (#sensors-1835958) entitled “Planar junctionless field-effect transistor for detecting biomolecular interactions” submitted to Sensors. In this article, the authors proposed a novel design of planar junction-less FET toward highly sensitive and multiplexed immunosensors for peptidomics and proteomics. The proposed FET design is a good candidate of the immunosensing platforms owing to its simplicity, chemical-stimuli responsiveness, and long-term operation stability. In addition, the basic performances of the device were well-characterized with sufficient data. However, I have several concerns with this research from the viewpoints of materials science and analytical chemistry. Thus, I would like to recommend the acceptance of this article only after minor revisions. The critical comments are as follows. 
1) To verify the surface activation of the device from the various aspects, the addition of elemental analysis for the APTES-immobilized surface utilizing XPS, SIMS, or ATR-FTIR measurements is required.

2) This research centered on the investigation of the basic performance in a new-designed device setup, whereas the authors should perform any biosensing experiment on the planar junction-less FET because the manuscript title claims the following sentence: “for detecting biomolecular interactions”. This is a critical point for the demonstration of the feasibility of the fabricated sensor device.

Reviewer 3 Report

The article with title: “Planar junctionless field-effect transistor for detecting bio-molecular interactions” deals with field-effect transistors as pH sensors employing SiO2 as gate oxide with a in-depth description of materials, working mechanisms as well as simulations. The article is well written, and certainly suitable for publications in sensors.

I do have a question or curiosity regrding the wet grown SiO2. During dry oxidation, the wafer is placed in a pure oxygen gas environment and the chemical reaction which ensues is between the solid silicon atoms (Si) on the surface of the wafer and the approaching oxide gas order to achieve a desired thickness. The oxide films resulting from a dry oxidation process have a better quality than those grown in a wet environment, which makes them more desirable when high quality oxides are needed. During wet oxidation, on the other hand, the silicon wafer is placed into an atmosphere of water vapor (H2O) and the ensuing chemical reaction is between the water vapor molecules and the solid silicon atoms (Si) on the surface of the wafer, with hydrogen gas (H2) released as a byproduct. Wet thermal oxide isn’t really designed for insulation/isolation and silicon is not oxidized at the same rate in each crystalline direction and the crystal orientation of the wafer plays a role in determining the dielectric properties of the oxide (permittivity for instance). Can the author comment on that, or how this might affect the overall device performance?

SiO2 and other similar oxides have been used previousley as pH sensing materials in FET architectures and this should be referenced. (https://doi.org/10.1021/nn5064216, https://doi.org/10.1002/aelm.201800381, https://doi.org/10.1080/10643389.2020.1843312).
